# Natural Killer Cells in Immunotherapy: Are We Nearly There?

**DOI:** 10.3390/cancers12113139

**Published:** 2020-10-27

**Authors:** Mireia Bachiller, Anthony M. Battram, Lorena Perez-Amill, Beatriz Martín-Antonio

**Affiliations:** Department of Hematology, Hospital Clinic, IDIBAPS, 08036 Barcelona, Spain; mbachiller@clinic.cat (M.B.); battram@clinic.cat (A.M.B.); loperez@clinic.cat (L.P.-A.)

**Keywords:** NK cells, immunotherapy, cancer, microbial infections

## Abstract

**Simple Summary:**

Here, we review the last pre-clinical and clinical studies published in the last five years where natural killer (NK) cells have been administered as an immunotherapy option for the treatment of cancer patients. We describe studies administering NK cells alone and in combination with monoclonal antibodies that either promote antibody-dependent cell cytotoxicity or block immune checkpoint receptors. We review the use of genetically modified NK cells including chimeric antigen receptor (CAR)-modified NK cells and other modifications that can enhance the anti-tumor activity of NK cells. Moreover, we describe studies related to the antimicrobial activity of NK cells as we believe they demonstrate important lessons that we can learn and apply to improve the anti-tumor activity of NK cells. All these studies are described with the aim to find tips to improve the success of NK cells as an immunotherapy option in cancer patients.

**Abstract:**

Natural killer (NK) cells are potent anti-tumor and anti-microbial cells of our innate immune system. They are equipped with a vast array of receptors that recognize tumor cells and other pathogens. The innate immune activity of NK cells develops faster than the adaptive one performed by T cells, and studies suggest an important immunoregulatory role for each population against the other. The association, observed in acute myeloid leukemia patients receiving haploidentical killer-immunoglobulin-like-receptor-mismatched NK cells, with induction of complete remission was the determinant to begin an increasing number of clinical studies administering NK cells for the treatment of cancer patients. Unfortunately, even though transfused NK cells demonstrated safety, their observed efficacy was poor. In recent years, novel studies have emerged, combining NK cells with other immunotherapeutic agents, such as monoclonal antibodies, which might improve clinical efficacy. Moreover, genetically-modified NK cells aimed at arming NK cells with better efficacy and persistence have appeared as another option. Here, we review novel pre-clinical and clinical studies published in the last five years administering NK cells as a monotherapy and combined with other agents, and we also review chimeric antigen receptor-modified NK cells for the treatment of cancer patients. We then describe studies regarding the role of NK cells as anti-microbial effectors, as lessons that we could learn and apply in immunotherapy applications of NK cells; these studies highlight an important immunoregulatory role performed between T cells and NK cells that should be considered when designing immunotherapeutic strategies. Lastly, we highlight novel strategies that could be combined with NK cell immunotherapy to improve their targeting, activity, and persistence.

## 1. The Potential of Natural Killer Cells

Natural killer (NK) cells have been recognized as potent anti-tumor and anti-microbial cells of the innate immune system. In peripheral blood, there are two main populations of NK cells, where 90% of them are CD56^low^ and CD16^high^, and are considered the mature and cytotoxic subpopulation [1], and which also express T-bet^high^ and Eomes^low^. In contrast, the remaining 10% are CD56^high^, CD25+, and CD16^low^, exhibit robust cytokine production, are less mature, less cytotoxic, and express T-bet^high^, Eomes^high^ [2].

The anti-tumor properties of NK cells have attracted a high level of interest in biomedicine. In the 1980s, several studies reported a higher incidence of cancers in individuals with defective NK cell function caused by genetic disorders, such as Chédiak–Higashi syndrome and X-linked lymphoproliferative syndrome [3,4]. During the same period, increased tumor growth and metastasis were described in mutant mice with impaired NK cell activity [5]. Impaired NK cells or NK cell deficiency were associated, not only with recurrent virus infections, but also with an increased incidence of various types of cancer [6].

As opposed to immune T cells that require a considerable length of time to acquire cytolytic activity, NK cells are “ready to kill”, and their activity is observed at earlier time points (within one hour) than in T cells. Moreover, the vast array of activating and inhibitory receptors on their surface equips NK cells with the capacity to recognize and kill a high variety of targets. These important features of NK cells made them the focus of attention in hemato-oncology, and led to the first evidence of their clinical benefit by Velardi and colleagues in 2002 [7]. They observed that acute myeloid leukemia (AML) patients who received T cell-depleted haploidentical allogeneic stem cell transplantation (allo-SCT), with a mismatch between the inhibitory killer-cell immunoglobulin-like (KIR) receptor in NK cells and the human leukocyte antigen (HLA)-I of the patient, experienced lower rates of relapse, suggesting that donor-derived NK cells were mediating an alloantigen-specific response against AML blasts, without causing graft versus host disease (GVHD). Importantly, this KIR-HLA mismatch, which can also occur when there is HLA-I down-regulation in tumor cells, activates NK cells after allo-SCT, leading to lysis of leukemia blasts, recipient dendritic cells, and recipient T cells, which translates into a reduction of relapse, prevention of GVHD, and avoidance of graft rejection, respectively [8,9]. These findings led to the adoptive cell transfer of in vitro-activated haploidentical KIR-mismatched NK cells into patients with AML. In these initial studies, two different conditioning regimens were tested, demonstrating that the more intense, high cyclophosphamide/fludarabine regimen resulted in a marked rise in endogenous IL-15, expansion of donor NK cells, and induction of complete hematologic remission in 26% of poor-prognosis patients with AML [10]. Since then, a high number of clinical trials have started to administer NK cells in patients, not only with AML, but also with other hematological malignancies and solid tumors.

In 2017, we reviewed published clinical studies administering NK cells for the treatment of hematological and solid tumors. Results up to that date suggested that allogeneic NK cells only showed a clear benefit when infused as a consolidation therapy in AML patients. Unfortunately, in refractory cancer patients with non-myeloid malignancies, NK cells were not successful [11]. Now, we review novel pre-clinical and clinical studies published in the last five years administering both NK cells and chimeric antigen receptor (CAR)-modified NK cells for the treatment of cancer patients. We also review combinatorial treatments with NK cells and other immunotherapy agents, such as monoclonal antibodies, and we suggest strategies to improve the activity of NK cells. Lastly, we present studies regarding the role of NK cells as anti-microbial effectors in the context of virus, bacteria, parasites, and fungus infections, as we believe that these studies are lessons to be learned, and to be incorporated into the use of NK cells as immunotherapy in cancer.

## 2. NK Cells as a Single Immunotherapy Option

The allogeneic origin of NK cells in immunotherapy strengthens the idea of an “off-the-shelf” product, because they can be available at any time. This is one of the main benefits of NK cells above the use of autologous T lymphocytes. In the last five years, many clinical trials administering NK cells have been started. However, many of them are either still recruiting patients or the results are not available yet. The most relevant studies are mentioned here and summarized in Table 1.

Clinical studies have used different NK cell sources, which include cord blood-derived NK cells (CB-NK) [12,13], peripheral blood NK cells (PB-NK) [10], NK cells derived from human induced pluripotent stem cells (iPSC-NK) [14], or NK cells derived from clonal cell lines, such as NK-92. Although NK-92 is dependent on IL-2, and cells die within 72 h if they lack the cytokine [15], in terms of safety, it has to be irradiated prior to infusion in patients, which can limit its therapeutic efficacy [16]. Regarding activation and expansion of NK cells, most protocols use cytokines such as IL-2, IL-12, IL-15, IL-18, and IL-21. Each cytokine impacts NK cell maturation, proliferation, survival, and distribution differently (reviewed in [17]). IL-15 has appeared as an important cytokine that increases the anti-tumor response of CD56^bright^ NK cells [18]. However, a disparity of opinions have emerged, as recently it was demonstrated that continuous in vitro exposure of NK cells to IL-15 leads to NK cell exhaustion [19]. Moreover, a clinical study in patients reported severe GVHD in cancer patients receiving allogeneic NK cells pre-activated in vitro with IL-15 and 4-1BBL and given HLA-matched T cell-depleted allogeneic hematopoietic stem cell transplants. GVHD was associated with higher donor CD3 chimerism, suggesting that NK cells might not be responsible for the GVHD development [20]. Bachanova et al. performed a phase II clinical trial in patients with refractory non-Hodgkin lymphoma (NHL), who received haploidentical NK cells with anti-CD20 antibody rituximab and IL-2 (NCT01181258) [21]. This study demonstrated safety without GVHD, cytokine release syndrome (CRS), or neurotoxicity, and the responding patients had lower levels of regulatory T (T-reg) cells and myeloid-derived suppressor cells (MDSCs) at baseline than non-responding patients. Importantly, endogenous IL-15 levels were higher in responders than non-responding patients at the day of NK cell infusion [21]. Moreover, although cytokine therapy can augment in vitro NK cell anti-tumor activity, the same approach in vivo may be limited by the systemic toxicity of cytokines [22]. In this regard, there is an ongoing clinical study evaluating the administration of haploidentical NK cells with the addition of subcutaneous IL-15 in AML patients (NCT03050216). Of interest, novel studies with CAR-NK cells include the addition of IL-15 secretion in the CAR construct [23], which avoids the administration of cytokines and the associated toxicities.

The addition of exogenous cytokines for NK in vitro expansion can be complemented or substituted by the use of artificial feeder cells, which provide a continuous source of cytokines for NK cells [12], obtaining a high number of NK cells to treat patients [12,13]. To take advantage of this approach in vivo, Chen et al. designed a chimeric protein with NKG2D and IL-15 that would bind to MICA on tumor cells and would trans-present IL-15 to NK cells. This strategy enhanced NK cell recruitment to tumor sites in mouse models of gastric cancer and melanoma, resulting in slowed tumor growth [24].

In recent years, some novelties introduced in the clinical protocols to expand NK cells have included the use of a combination of cytokines to obtain “memory-like” NK cells. Thus, NK cells expanded with IL-12, IL-15, and IL-18 induce a memory-like phenotype, with increased IFNγ production, expression of the high affinity IL-2 receptor, and cytotoxicity against AML blasts [25,26]. In mouse models of AML and melanoma, it was demonstrated that memory-like NK cells proliferate in vivo and exhibit increased effector function, resulting in enhanced tumor clearance and improved survival [25,26]. Moreover, in a phase I clinical trial, adoptively transferred memory-like NK cells proliferated and expanded in AML patients during the first week and demonstrated ex vivo responses against leukemia targets. In nine evaluable patients, an objective response (OR) of 55% and a complete remission (CR) rate of 45% was observed [25]. Even though the number of patients was small, results were suggested to be better than those obtained by Bachanova et al. in 57 AML patients treated with haploidentical NK cells and IL-2 administration, where they obtained 21% of CR. However, in the same study by Bachanova et al., they included a cohort who received IL-2-diphtheria fusion protein (IL-2DT) that depletes T-reg cells, and which significantly improved CR to 53% [27].

Khatua et al. [28] performed a phase I clinical trial of 12 pediatric patients with brain tumors who received intraventricular infusion of autologous NK cells expanded with feeder cells. Nine evaluable patients were treated, receiving up to three infusions weekly. Safety of 112 intraventricular infusions of NK cells was achieved in all nine patients. There were no dose-limiting toxicities. However, despite the high amount of NK cells administered, all patients showed progressive disease (PD), except one patient who showed stable disease (SD) for one month. Of note, frequent infusions of NK cells resulted in cerebrospinal fluid pleocytosis, with the presence of NK cells.

Björklund et al. evaluated the administration of IL-2-activated haploidentical NK cells in primary relapsed/refractory (R/R) high-risk myelodysplastic syndrome (MDS), secondary AML (MDS/AML), and de novo AML patients. A total of 16 patients were treated and NK cells were well tolerated. Six patients (37.5%) achieved ORs with CR, marrow CR, or partial response (PR). However, five patients proceeded to allo-SCT afterwards. Three patients were still free from disease three years after treatment. All evaluable patients with OR had detectable donor NK cells at days 7/14 following infusion. Responding patients displayed less pronounced activation of CD8+ T cells and lower levels of inflammatory cytokines following NK cell infusion. All patients displayed increased frequencies of activated T-reg cells of recipient origin following NK cell therapy [29]. These findings suggest some type of immunoregulatory activity performed between both T cells and NK cells, as has been observed for the microbial infections that we will discuss in Section 5.

Another phase II study in relapsed or progressive AML or MDS was performed, in which patients were treated with haploidentical NK cells after cyclophosphamide-based lymphodepletion following allo-SCT. A total of eight patients were treated with a median of 10.6 × 10^6^ NK cells/kg and six doses of IL-2 every other day. Safety was demonstrated without incidence of GVHD [30]. As in the study by Björklund et al. [29], 37.5% of patients achieved OR. Two patients achieved CR (one patient with AML and one patient with MDS); however, they relapsed at 1.7 and 1.8 months. The median overall survival (OS) was 12.9 months. Of note, in this study, NK cells were not detected after infusion [30].

Yang et al. performed a phase I study (NCT01212341) evaluating repetitive administrations of allogeneic expanded NK cells from random unrelated healthy donors (MG4101) into patients with lymphoma or refractory solid tumors. Safety was demonstrated for the maximum dose (3 × 10^7^ cells/kg, triple infusion). Of 17 evaluable patients, 47.1% showed SD and 52.9% showed PD. Of interest, it was observed that MG4101 reduced T-reg cells and MDSCs, increased NKG2D expression on CD8+ T cells, and upregulated the chemokines that recruit T cells [31].

In solid tumors, there are ongoing studies evaluating intraperitoneal administration of CB-NK to treat recurrent ovarian carcinoma (NCT03539406) or oral cavity carcinoma in pediatric patients (NCT03420963). However, results are not available yet.cancers-12-03139-t001_Table 1Table 1Clinical results in the last 5 years administrating natural killer (NK) cells.NCT Number. Phase. Investigator. ReferenceSource of NK and Method of ExpansionStage of Disease and Number of PatientsClinical OutcomeNCT01181258. Phase II. Bachanova, V. et al. [21]Allogeneic PB-NK + IL-2 (1000 IU/mL)R/R NHL or CLL CD20^+^. 14 evaluable patients.4/14 OR at 2 months (28%).2/24 CR for 9 months (14%)NCT01898793. Phase I/II. Romee, R. et al. [25]Haploidentical PB-NK + 12–16 h: IL-15, IL-12 and IL-18. 3 doses: 0.5, 1 and 10^6^ NK/KgR/R AML (*n* = 13, 9 evaluable).Well tolerated, no GvHD. OR: 55%CR: 45%NCT02271711. Phase I. Khatua, S. et al. [28]Autologous PB-NK + K562-mbIL-21R/R brain tumor: medulloblastoma (*n* = 5) and ependymoma (*n* = 4) in pediatric patientsSD: 11.1%PD: 88.9%NCT00526292. Phase II. Shaffer, B. C. et al. [30]Haploidentical PB-NK. 6 doses of IL-2 in patients every other day.AML (*n* = 6) and MDS (*n* = 2)No GvHDPR: 37.5%CR: 25%Median OS = 12.9 monthsEudraCT number 2011-003181-32. Bjorklund, A. T. et al. [29]Allogeneic PBNK + IL-2 (1000 IU/mL)R/R or high-risk MDS (*n* = 5), MDS-AML (*n* = 9) or de novo AML (*n* = 3). 16 evaluable.OR: 37.5% and SD: 12.5%5/16 underwent allo-SCT. Of these in 3/16, DFS > 3 yearsNCT01212341. Phase I. Tae Min Kim [31]Allogeneic PB-NK.14 days of expansion with irradiated auto-PBMCs, OKT3 +IL-2 (500 IU/mL) every other dayLymphoma (*n* = 2) and solid tumor (*n* = 19). 17 evaluableNo GvHD, no severe toxicities. 47.1% SD, 52.9% PD, median PFS in SD patients of 4 monthsR/R: relapsed/refractory; OR: objective response; SD: stable disease; PR: partial response; PD: progressive disease; CR: complete response; GvHD: graft-versus-host disease; NE: not evaluable; MLFS: morphologic leukemia-free state; allo-SCT: allogeneic stem cell transplantation; OS: overall survival; PFS: progression free survival.

## 3. Use of NK Cells in Combination with Monoclonal Antibodies

The little success observed administering NK cells as a single therapy has led to their combination with other immunotherapeutic tools to improve their efficacy. Monoclonal antibodies bind to surface targeted molecules expressed on cancer cells, and owe their mechanism of action partially to NK cell-mediated antibody-dependent cell cytotoxicity (ADCC). NK cells express CD16 (FcRIII receptor), which is key to ADCC. Thus, among the studies being performed, there is a phase I clinical trial (NCT02030561) administering NK cells after trastuzumab (anti-HER2) treatment [32]. Autologous NK cells were expanded with K562-mb15-41BBL artificial antigen presenting cells for 10 days. Prior to infusion, NK cells expressed high levels of CD16, which was down-regulated once infused in the patient. Nine patients with HER2+ breast or gastric cancer received trastuzumab and subcutaneous IL-2 the day before NK cell infusion. Subsequently, patients received IL-2 three times a week and three more cycles of trastuzumab. The combination treatment was well tolerated but no OR was observed. In total, 66.6% of patients achieved SD for >6 months and 11% accomplished PR. The results obtained set the basis for a future phase II trial with 20 patients [32]. Another clinical study, analyzed the combination of NK cells with cetuximab (anti-EGFR) in 54 patients with advanced non-small cell lung cancer. Patients were randomized to receive either cetuximab plus NK cells or cetuximab alone. Safety was demonstrated in both groups, and the group receiving NK cells presented lower levels of carcinoembryonic antigen, neuron specific enolase, and circulating tumor cells than before treatment. Moreover, in comparison to cetuximab alone, they had a significant improvement in immune function and quality of life, and survived longer (median PFS: 6 months vs. 4.5 months; median OS: 9.5 months vs. 7.5 months) demonstrating a beneficial effect of combining NK cells with antibodies that promote ADCC of NK cells [33].

The use of immune checkpoint inhibitors is another area of interest for NK cell combination therapies. NK cells express a wide variety of inhibitory receptors, including NKG2A, KIRs (such as KIR3DL2), PD-1, TIGIT, TIM-3, and LAG-3. Monoclonal antibodies that block inhibitory receptors have shown great efficacy in enhancing T cell/CAR-T cell anti-tumor activity, and a number of clinical trials are currently ongoing that are testing some of these checkpoint blockade molecules as a means to enhance endogenous T cell and NK cell activity (reviewed in [34]).

For NKG2A, Andre P. et al. [35] demonstrated that blocking of NKG2A with monalizumab (a humanized anti-NKG2A antibody) enhanced NK cell anti-tumor activity in various tumor cells. Moreover, monalizumab promoted NK cell ADCC, as when combined with cetuximab and obinutuzumab (anti-CD20) it led to the amplified activation of NK cells with enhanced ADCC. Thus, monalizumab can amplify the beneficial effects of other treatments which promote ADCC. Interim results of a phase II trial (NCT02643550) of monalizumab plus cetuximab in previously treated patients with squamous cell carcinoma of the head and neck showed a 31% objective response rate. However, in this study, patients did not receive NK cells [35]. In chronic lymphoid leukemia patients, tumor cells overexpress HLA-E, and NK cells overexpress NKG2A. Blocking NKG2A with monalizumab on CLL NK cells restored the cytotoxicity ability of NK cells against HLA-E-expressing targets, without impacting NK cell ADCC [36]. In addition, the blockading of NKG2A is an interesting approach that allows the enhancement of NK cell alloreactivity after haploidentical-SCT. In this setting, Roberto A. et al. characterized that after haploidentical-SCT there is a transient and predominant expansion of an unconventional NK cell population, characterized by NKp46^-low^/CD56^dim^/CD16^-^ with high levels of CD94/NKG2A. This expansion starts from the second week following haploidentical-SCT. While present at low frequency in healthy donors, this unconventional NK cell population express high levels of NKG2D, NKp30, Granzyme-B, and Perforin, but displays defective cytotoxicity that could be reversed by blocking CD94/NKG2A [37].

Inhibitory KIRs are inhibited with lirilumab. Kohrt et al., using a KIR transgenic murine model, demonstrated that the blockade of inhibitory KIRs with lirilumab augments NK cell cytotoxicity. Moreover, in combination with rituximab (anti-CD20), anti-KIR treatment induced enhanced NK-cell-mediated, rituximab-dependent cytotoxicity in murine lymphoma models. These results support a therapeutic strategy combining rituximab and KIR blockade through lirilumab [38]. Lirilumab has demonstrated safety in cancer patients, being administered alone and in combinations with other drugs [39,40]. However, clinical results in combination with NK cells are not available yet.

Regarding PD-1, it was observed that mature CD56^dim^ NK cells display high levels of PD-1, being characterized by a NKG2A^−^ KIR^+^ CD57^+^ phenotype. This PD1+ NK cell population showed reduced functional activity, and is present in approximately one fourth of healthy subjects. Interestingly, these donors are always serologically positive for human cytomegalovirus. These NK cells might have a preferential expansion in tumor environments. Therefore, authors suggested that their blockade with anti-PD1 could promote NK cell-mediated cytotoxicity [41].

Based on these results, different studies are now considering the application of checkpoint inhibitors to improve NK cell function in adoptive therapy, particularly those that target PD-1 and its ligand, PD-L1. NK cells can promote the expression of PD-L1 in tumor cells through secretion of IFNγ, which generates an immunosuppressive tumor microenvironment that impedes NK cell activity [42]. Therefore, it is expected that the use of PD-L1 inhibitors could potentiate the activity of NK cells. On the other hand, PD-1 antibodies can bind to PD-1 molecules on NK cells’ surface to prevent their depletion [43]. In the only clinical study combining NK cell therapy and a checkpoint inhibitor that currently has published results, Lin et al. performed a clinical trial in 109 patients with non-small cell lung cancer (NSCLC) that were randomly treated with pembrolizumab (anti-PD-1) plus NK cells or given pembrolizumab alone. Higher OS and PFS rates were achieved with the combination therapy compared to those obtained by the group in monotherapy (15.5 and 6.5 months vs. 13.3 and 4.3 months, respectively). Furthermore, better median OS was observed, comparing patients who received more than one infusion vs. a single infusion (18.5 months vs. 13.5 months) [44]. Many other clinical studies are currently ongoing. In one pilot study (NCT03958097), the combination of sintilimab (anti-PD-1), which is currently approved in China for R/R Hodgkin’s lymphoma [45], is being tested as a dual therapy with autologous NK cells. No results have been posted yet.

Suppressor cells represent another major obstacle for optimal NK cell function in the tumor microenvironment. Tumor-associated neutrophils, which repress NK cell cytotoxicity through the PD-1/PD-L1 axis [46], and MDSCs are both potential target cells for checkpoint therapy. Indeed, CAR-NK cells that target PD-L1-expressing cells efficiently eliminate MDSCs in vitro [47]. Clinical trials that combine NK cells with monoclonal antibodies are summarized in Table 2.

Moreover, the efficacy of combining checkpoint blockade with other drugs to improve NK cell tumor clearance is beginning to be explored as an attractive approach that has enhanced NK cell efficacy. These combinations are discussed below in Section 6.

## 4. Chimeric Antigen Receptor (CAR)-Modified NK Cells

The success achieved with the administration of genetically-modified CAR-T cells in hematological malignancies [48] has led to its adaptation to try to improve the efficacy of NK cells. It is expected that the use of genetically modified CAR-NK cells would avoid the low persistence and low efficacy of NK cells, and the failure to migrate to the tumor site [49]. Moreover, unlike CAR-T cells, CAR-NK cells would also present spontaneous cytotoxic activity, independently of target antigen [50]. In addition, CAR-NK cells can be obtained from different sources of NK cells, such as cell lines, PB-NK, CB-NK, human embryonic stem cells (hESCs), or iPCSs [51,52,53]. The selection of each source should be defined based on the type of disease to be treated, and also depending on the urgency of the treatment. Thus, in chronic diseases where CAR-NK cells should persist as long as possible, it is expected that CB-NK or iPSC-NK would be better, as they are more immature with a longer lifespan [54]. Otherwise, if the patient must be treated immediately, CAR-NK cells derived from irradiated cell lines could be ready at any time, off-the-shelf [53].

Even though CAR-NK cells can be a very suitable source of immune cells for cell immunotherapy, results so far have demonstrated that their efficacy is still far from that obtained with CAR-T cells. Pre-clinical studies evidenced that NK cell lines appear to be the major source for obtaining CAR-NK cells, due to their easier transduction rates than primary NK cells [53,55]. There are several NK cell lines available, such as NK-92, HANK-1, NK-YS, NKT, or YTS [53], with the NK-92 cell line the most common source used, since it has shown remarkable cytotoxicity vs. tumor cell lines [51]. On the contrary, the major disadvantage of using NK cell lines is that they must be irradiated before administration in patients, which means their therapeutic window is limited. Pre-clinical studies with CAR-NK cells obtained from NK cell lines have shown an increased anti-tumor efficacy. These studies are summarized in Table 3.

Clinical studies published administering CAR-NK cells derived from cell lines have not been optimal. The only published study was performed in patients with R/R AML who received CAR NK-92 cells directed against CD33 [69]. The CAR was a third generation construct with CD28 and 4-1BB co-stimulatory domains. Three AML patients were treated with 5 × 10^6^ of CAR-NK cells. The first and second patients received three CAR-NK doses (3 × 10^8^, 6 × 10^8^, and 1 × 10^9^ cells) on days 1, 3, and 5, respectively. CAR-NK cells were found in patients one week after the administration. One month after CAR-NK treatment, the first patient treated showed clearance of blasts in the bone marrow. However, four months later, the patient relapsed with 76% of blasts in the bone marrow. The second patient presented 75% of blasts one month after therapy. The third patient, even though he received much higher doses of CAR-NK cells (1 × 10^9^, 3 × 10^9^, and 5 × 10^9^, on days 1, 4, and 7, respectively), exhibited no response to treatment. Of note, none of the patients presented higher CRS than grade 1 or any other side effects [69].

On the other hand, there are fewer studies using CAR-NK cells obtained from primary NK cells. The first pre-clinical study was published in 2005 by Dr. Campana’s group. They modified PB-NK to express an anti-CD19-BB-z CAR to treat leukemias and lymphomas. CAR-NK cells were expanded using K562-based artificial antigen presenting cells, which yielded a median greater than 1000-fold expansion of NK cells at 3 weeks of culture and an increased anti-tumor efficacy in vitro was achieved [70].

An additional consideration for CAR-NK cells is the choice of signaling domains to incorporate into the CAR to obtain optimal cell stimulation. In this regard, studies in CAR-T cells comparing 4-1BB and CD28 demonstrated more robust signaling for CD28 than for 4-1BB. However CD28 CARs displayed increased T cell dysfunction [71]. Most CARs used in NK cells were originally designed for T cells, and due to the differences in T cell and NK cell signaling, they are likely to be suboptimal when applied in NK cells [64,71,72].

Recently, studies have emerged creating new construct designs specifically for NK cells. In 2013, Chang et al. designed a CAR composed of NKG2D receptor with the adaptor protein DAP10 as the co-stimulatory domain, and demonstrating increased anti-tumor activity against different types of tumors in vitro and in a murine model of osteosarcoma [73]. On the other hand, CAR-NK cells that contain DAP10 have had mixed results, despite some success when used in combination with NKG2D, an endogenous partner receptor of DAP10 [72]. A first generation CAR that used the adaptor protein DAP12 in place of CD3ζ displayed promising anti-tumor cytotoxicity in vitro and in vivo [64]. It was identified that exchanging the classic transmembrane and co-stimulatory domains used for CAR-T cells for other domains more likely to be expressed in NK cells could mediate stronger CAR-NK activation and cytotoxicity effect [67,74]. Thus, Li et al. [74] compared different second and third generation CAR constructs in NK cells derived from iPSC cells, and demonstrated that CARs with NKG2D as transmembrane domain and 2B4 as co-stimulatory domain were crucial for activating endogenous signaling and conferring strong NK cell anti-tumor cytotoxicity, compared to CAR-NK with T cell domains. Moreover, similar in vivo activity of CAR-NK cells compared to CAR-T cells was observed in a xenograft mouse model of ovarian cancer.

Other variants include a CAR designed by Zhu et al. [75] with a mutant variant of CD16 that is not cleaved by the disintegrin and metalloproteinase-17. These CAR-NK cells, combined with rituximab, demonstrated superior anti-tumor activity in in vivo models of B-acute lymphoblastic leukemia (B-ALL). Moreover, Liu et al. [23] demonstrated that the addition of IL-15 in the construct of an anti-CD19 CAR increases the persistence and anti-tumor activity of CB-NK. Presence of CAR-NK cells was detected up to 68 days post infusion in a murine lymphoma model. Of note, this CAR incorporated an inducible Caspase-9 system to eliminate NK cells in vivo in case it was necessary due to the development of severe toxicities [23]. The most relevant pre-clinical studies with CAR-NK cells obtained from primary NK cells are summarized in Table 4.

Despite the efficacy of CAR-NK cells in pre-clinical studies, there have been very few clinical trials performed with CAR-NK cells derived from primary NK cells. The first ones were performed at St. Jude Children’s Research Hospital, Memphis (NCT00995137) and the National University Hospital, Singapore (NCT01974479). Both clinical studies administered haploidentical CAR-NK cells against CD19 for the treatment of B-ALL. Both studies used feeder cells to increase the expansion of CAR-NK cells. Results are not available yet. The CAR designed by Liu et al. [23] was used in a clinical trial to treat 11 NHL or CLL patients (NCT03056339). No CRS or neurotoxicity was observed in any patient. From 11 patients, 8 had a response, and 7 a complete response. CAR-NK cells were detectable for at least 12 months. However, as most patients were treated shortly after CAR-NK administration with additional treatments, such as rituximab, it is difficult to evaluate the efficacy of CAR-NK cells [81].

## 5. Lessons Learned from the Antimicrobial Properties of NK Cells

Although the adoptive transfer of NK cells or CAR-NK cells has shown exciting promise as a cancer therapy, their immunotherapeutic efficacy in clinical trials has often been limited. Importantly, many of the studies found that NK cells did not persist for a long time [13,30], and that in some cases, responding patients had lower levels of CD8+ T cells [29], suggesting an immunoregulatory effect of T cells over NK cells. In this regard, the important immunoregulatory role performed by both populations against each other is well-known [82,83]. Specifically, studies with T cells and NK cells in the context of viral infections have demonstrated that NK cells eliminate T cells [84], but also that T cells can eliminate NK cells [85] to avoid an exacerbated inflammatory response with fatal consequences. Moreover, studies in viral infections with NK cells demonstrated the development of NK cell memory and markers associated with this event, which might be useful for immunotherapy. Thus, a deeper insight into the antimicrobial properties of NK cells could help to improve NK cell immunotherapy strategies for cancer patients.

### 5.1. NK Cell Responses and Development of Memory-Like NK Cells after Viral Infections

Different studies demonstrated that NK cells develop a “memory” like phenotype after viral infection (reviewed in [86]). This memory feature of NK cells was mostly studied in the context of cytomegalovirus (CMV) infection. In 2009, Sun et al. [87] demonstrated in mice models that after CMV infection, NK cells show a similar response to memory T cells, with a proliferation phase followed by a contraction phase, and that afterwards these NK cells reside in lymphoid and non-lymphoid organs for several months. Moreover, adoptive transfer of “memory” NK cells into naive animals followed by viral challenge results in a robust secondary expansion and protective immunity. Afterwards, this event was also observed in humans during CMV infection, where a higher expansion of NK cells expressing NKG2C and CD57 was detected in a small proportion of CMV+ individuals. In addition, this population remained higher in CMV-seropositive healthy individuals than in seronegative individuals. Thus, this study proposed CD57 as a possible memory NK cell marker [88]. Additional studies in human samples also confirmed the expansion of NKG2C+ NK cells after CMV infection, and that these NK cells also express the activating KIRs, KIR2DS4, KIR2DS2, or KIR3DS1 [89].

The existence of memory-like NK cell development after viral infection was confirmed for other viruses, such as Chikungunya virus, which also induces a preferential expansion of NKG2C+ NK cells. However, in this study, all individuals were CMV+; thus, it cannot be excluded that this preferential expansion was due to CMV [90]. Regarding Epstein–Barr virus (EBV) infection, different studies concluded contradictory results. In children, CMV and EBV infections can occur in close temporal proximity. Therefore, Saghafian-Hedengren et al. [91] analyzed whether EBV infection may impact CMV immunization. They confirmed the increased frequency of NKG2C+ NK cells in CMV+ children and among those, they observed that EBV and CMV co-infection led to higher proportions of NKG2C+ NK cells than in the absence of EBV infection. However, afterwards, another study performed in university students confirmed the presence of NKG2C+ NK cells after CMV infection, but not the increase of this population with EBV infection [92].

Additional studies in murine models for other viruses have confirmed the existence of NK cell antiviral activity, with development of short-time memory in the absence of T cells in models of genital herpes simplex virus type-2 infection [93], vaccinia virus infection [94], and influenza virus [95]. In addition, in rhesus macaques, it was demonstrated that NK cell activation and cytotoxicity against simian immunodeficiency virus peaks at one or two weeks post-infection [96].

NK cells have also been studied in the context of human immunodeficiency virus (HIV) infection, showing their association with efficient responses and long-term resistance to the virus. As such, HIV-infected long-term non-progressors and HIV controllers represent a population of interest to study their immune cells. This population exhibits peculiar phenotypic NK features, associated with high levels of activation of NK cells, where a role for NKp44 has been suggested [97]. Moreover, NK cell response to HIV has been studied in highly exposed seronegative individuals (HESN) who show natural resistance to HIV acquisition, despite repeated exposures. HIV-exposed intravenous drug users are a type of HESN who present increased CD69 activation marker and CD107a degranulation in their NK cells [98]. Furthermore, NK cells from HESN women exhibit differential receptor expression compared to NK cells from healthy donors. HESN NK cells present increased expression of NKG2A, NKp30, LILRB1, and CD16, and decreased expression of DNAM-1, CD94, Siglec-7, and NKp44. These NK cells have increased ADCC, which correlates with increased CD16 [99].

NK cell memory responses against varicella-zoster virus (VZV) have also been observed in humans. It was shown that large numbers of NK cells were recruited to sites of VZV skin test antigen challenge, even decades after initial exposure, demonstrating that development of NK cell memory in humans after viral infection is long-lived [100].

### 5.2. Coordinated Immune Response after Viral Infections

An important aspect that needs to be considered is the global and coordinated immune response performed by the different immune cell populations. In viral infections, the interaction of NK cells and T cells is of particular interest. Murine models of viral infection demonstrated that high viral loads lead to excessive T cell activation that induces fatal tissue damage and mortality [84]. In this scenario, NK cells present an important immunoregulatory role for T cells to maintain immune homeostasis and importantly limit damage to vital organs that can lead to death. Thus, in murine models of *lymphocytic choriomeningitis* virus, *Arenavirus*, *Pichinde* virus, and *Coronavirus* mouse hepatitis virus a three-way interaction was demonstrated between NK cells, CD4+ T cells, and CD8+ T cells. At high viral loads, and to prevent fatal pathology, activated NK cells eliminated activated CD4+ T cells, impacting negatively on CD8+ T cell function and exhaustion, while enabling viral persistence. However, at a medium viral load, T cells proliferated with production of inflammatory cytokines and concomitant pathology that was not inhibited by NK cells, and was similarly detrimental for the host [84]. The inhibition of T cells by NK cells has also been observed for other viruses, such as hepatitis B virus [101], and murine CMV, where even though this T cell elimination prolonged the chronicity of infection it also prevented autoimmunity [102]. Whereas this NK cell activity is performed to prevent fatal pathology, it has been suggested that in the case of patients with a chronic viral infection, NK cell depletion might restore the anti-viral immune function, enhancing viral control [103]. Moreover, the preferential elimination of different T cell subsets by NK cells has been analyzed, and it was found that T-reg, Th17, and Th2 cells were more sensitive than Th0 and Th1 cells [104]. Of interest, not only NK cells eliminate T cells to prevent fatal tissue pathology, but T-reg cells can also eliminate NK cells [85] to avoid an exacerbated inflammatory response with fatal consequences. Of note, after bone marrow transplantation T-reg cell reconstitution takes only 1 to 6 weeks [105]. This observation might be considered when administering NK cells in cancer patients.

Another crucial immunoregulatory role of NK cells for T cells has been observed by us in the context of tumor cells, but in this case, NK cells stimulate T cell activity. We found that NK cells, by releasing pro-inflammatory molecules, such as histones, promote the anti-tumor activity of T cells by accelerating T cell/tumor cell cluster formation [106]. All these studies showing the immunoregulatory role performed between NK cells and T cells should be considered in immunotherapy to find a balance between both immune populations, as both are required to eliminate tumor cells in an efficient manner.

### 5.3. Collaboration of NK Cells and T Cells after Parasitic Infections

In the context of parasitic infections, NK cells also play an important role modulating T cell responses. T cells are critical for the control of chronic toxoplasmosis caused by *Toxoplasma gondii* (*T. gondii*) infection, where parasite reactivation has been attributed to the development of immune exhaustion of parasite-specific CD4+ and CD8+ T cells [107]. At early stages of acute infection, and as a result of IL-12 signaling, NK cells produce IFNγ, which helps to control the parasite prior to T cell activation. However, at late time points, and during chronic *T. gondii* infection, CD8+ T cells become exhausted and are then removed by NK cells, resulting in parasite reactivation and death. The specific type of NK cells which are induced after chronic *T. gondii* infection do not produce IFNγ but do increase CD107a, and are enriched for NKp46, NKG2A, and KLRG1. Importantly, NK cell depletion or blockade of NKp46 rescued mice chronically-infected with *T. gondii* from death, thus demonstrating the fatal role of NK cells in this context [108]. These results show that NK cells have a specific timing to perform their beneficial activity of being accessory cells that constrain the viral infection at early time points, while T cells start to develop their response. This observation should be also contemplated in immunotherapy strategies for cancer patients, to find the perfect team between both immune cell populations.

An interesting phenomenon of plasticity for NK cells and innate lymphoid cells (ILCs) has been observed after parasite infections. Despite their resemblance, ILCs and NK cells were recently classified as different lineages [109], and both become activated under viral infection [110] and tumorigenesis [111]. Surprisingly, in certain environments, such as tumors, NK cells can be converted into ILC1-like cells that are unable to control local tumor growth and metastasis [112]. This plasticity phenomenon has also been observed after infection with *T. gondii*. Initially, both NK cells and ILC1s respond to this parasite. However, after *T. gondii* infection in mice, a large and permanent conversion of NK cells into ILC1-like cells occurs. These ILC1-like cells clear the infection and appear as a distinct population from both steady-state NK cells and ILC1s. They show lost expression of Eomes and gained expression of Ly6C, KLRG1, CX_3_CR1, and Nrp-1, retaining the ability to produce IFNγ, they do not produce TNFα, and are maintained in the absence of ongoing stimulus [113].

### 5.4. Coordinated Immune Response Against Bacteria for NK and T cells

A variety of emerging bacterial infectious diseases are transmitted through ticks, and include pathogenic species such as *Anaplasma spp*., *Ehrlichia spp*., and *Rickettsia spp*. (reviewed in [114]). In detail, *Anaplasma phagocytophilum* infection, which causes human granulocytic anaplasmosis, induces a proinflammatory disease state with innate immune cell activation. In murine infection models of this bacteria, a deficient immune response has been determined, not only for NK cells, but also for NKT cells and CD8+ T cells, due to decreased degranulation and IFNγ production [115].

A role for NK cells in the clearance of *Ehrlichia* infection, which causes potentially fatal human monocytic ehrlichiosis, has been observed. It was demonstrated that, compared to naïve mice, mice that overcame primary infection with *Ehrlichia muris* showed a better protection and survival following a secondary infection with highly virulent *Ixodes ovatus Ehrlichia*, but that this heightened memory response was abolished if NK cells were depleted in *E. muris*-primed mice. Moreover, the immunoregulatory role of NK cells, activating the adaptive immune response, was also observed here, as NK-depleted mice had a reduced number of *Ehrlichia*-specific memory CD4+ and CD8+ T cells, and a low titer of *Ehrlichia*-specific antibodies [116].

In murine models of *Rickettsia* infection, it was also suggested that the infection seems to be controlled initially by NK cells through IFNγ secretion, prior to the late response of CD8+ T cells. Moreover, similarly to virus infection, lack of NK cells caused serious tissue pathology after *Rickettsia* infection [117]. Infection with *Orientia tsustugamushi*, which causes scrub typhus, has been shown to correlate with an increase of peripheral blood CD69+ NK cells in patients. Moreover, NK cells produced a higher amount of IFNγ, and their levels correlated with severity of the disease [118].

It is important to highlight the balance required between clearance of the pathogen and lethal tissue damage due to an exacerbated immune response, a phenomenon observed both after viral [84] and bacterial infection [117]. In this regard, IL10 production after *Streptococcus pneumoniae* infection is important to avoid excessive inflammation of tissues and to improve host survival, even though there is lower bacterial dissemination in the absence of this cytokine [119]. Of interest, in a murine model of sub-lethal *Streptococcus pneumonia* infection, NK cells produced IL10, which restricted host protection [120], indicating again the immunoregulatory role of NK cells.

Different studies have analyzed the role of NK cells in response to *Mycobacterium bovis* bacille Calmette-Guérin (BCG) infection, demonstrating again the collaborative role between NK cells, T cells, and other cells of the immune system, and showing different timings for each cell subset. Thus, Esin, S. et al. [121] demonstrate that the NKp44 receptor is directly involved in the recognition of BCG, and other bacteria of the genus Mycobacterium, *Nocardia farcinica,* and *Pseudomonas aeruginosa*, by CD56^bright^ NK cells. Moreover, the infection of BCG was analyzed in two different models of infection. In the first, autologous monocytes/macrophages were infected with intracellular BCG, and in the second, BCG infection was extracellular. Whereas in response to intracellular BCG, CD4^+^ T cells were the main cell subset responding to infection, with a proliferation peak at 7 days, NK cells were involved in response to extracellular BCG, with peak IFNγ production at 24–30 h, and a peak of proliferation at 6 days [122]. In addition, Marcenaro, E. et al. [123] suggested that BCG might induce simultaneous activation of NK cells and antigen presenting cells via their “shared” toll-like-receptor-2, promoting efficient bidirectional NK cell-dendritic cell interactions, necessary for subsequent priming of CD4+ T cell responses.

### 5.5. NK Cells Alone and in Collaboration to Fight Fungal Infections

The incidence of invasive fungal infections is a relevant issue in cancer patients receiving either allo-SCT or organ transplantation, where fungal infections are associated with significant morbidity and mortality. Therefore, new options need to be considered to improve the outcome of these patients [124,125]. In this regard, clinical studies evaluating patients receiving allo-SCT demonstrated that those who suffer from invasive aspergillosis displayed insufficient NK cell recovery and lower ROS levels, and patients who were cured from invasive aspergillosis had higher NK cell counts and higher ROS production [126]. Moreover, patients undergoing solid organ transplantation who presented a higher mean NK cell count suffered less development of invasive fungal disease [127]. Of note, immunosuppressive treatments required in these patients, such as methylprednisolone and cyclosporine A have a negative impact on NK cell function and survival, suggesting that modifications of clinical protocols should be considered for NK cells to maintain activity [128]. Moreover, addition of antifungal agents at therapeutic doses to NK cells did not inhibit the functional activity of NK cells [129].

Fungal infections appear to be resolved not only by NK cells, but with the collaboration of the different subsets of immune cells. Thus, neutrophils were described years ago as the main immune mediators in fungal infections, through release of neutrophil extracellular traps (NETs) that contain antimicrobial proteins, such as histones and other pro-inflammatory molecules. Importantly, the release of pro-inflammatory molecules in these NETs has the ability to recruit other immune cells [130,131]. Of note, we have also observed that CB-NK [11] release histones with anti-fungal in vitro and in vivo activity against *Candida albicans* (unpublished data). Moreover, this immunomodulatory role for NK cell histones was also observed in a tumor context, where histones released by NK cells were able to recruit T cells to perform anti-tumor activity [106].

Additional immune cell populations cooperating in fungal infections also include CD8+ T cells. For example, in *Cryptococcus neoformans* infection in HIV patients, the antifungal activity of NK cells is constitutive and mainly mediated through perforin, whereas CD8+ T cells require prior activation and use granulysin [132]. Moreover, macrophages are also involved in NK cell activation during fungal infection. For instance, *Aspergillus fumigatus* infection induces the polarization of macrophages into pro-inflammatory M1 macrophages that secrete galectin-9, TNFα, and IL-18, and promote NK cell activation, as evidenced by enhanced CD69 expression and IFNγ secretion [133].

The mechanisms and receptors involved in NK cell anti-fungal activity have been extensively reviewed [134]. Most relevant studies indicate an important role for the family of NCR receptors in NK cells. Thus, NK cells recognize *Candida albicans* and *Candida neoformans* through NKp30, which further mediates killing of these fungi through phosphatidylinositol 3-kinasesignaling and perforin release. Moreover, HIV patients with defective antifungal activity have decreased NKp30 levels [135], where IL-12 restores NKp30 expression and fungal killing [136]. NKp46 receptor has been identified to recognize *Candida glabrata* [137], and CD56 interacts with *Aspergillus fumigatus,* and is involved in the NK cell anti-fungal activity [138].

Not only histones but also other pro-inflammatory molecules, such as heat shock proteins, present in neutrophil-derived NETs [131], also participate in the anti-fungal activity of NK cells [139]. In this regard, we also noticed that heat shock proteins are among the pro-inflammatory proteins released by NK cells when they encounter multiple myeloma cells [106]. Studies in fungal infections with NK cells have also demonstrated the important collaboration between all immune cell subsets, a finding that might be considered to create a balance in cancer immunotherapy that does not inhibit the activity of NK cells.

## 6. Improving NK Cell Activity

In addition to these lessons that we can learn from microbial infections and NK cells. There is a pressing need to improve current NK cell immunotherapies and develop novel NK cell-based treatments. Recent improvements have focused on more efficient targeting of NK cells to tumor cells, enhancing NK cell function, reducing NK cell suppression, and increasing NK cell persistence. In this section, we discuss some of the cutting-edge strategies being explored to achieve these aims. These strategies are summarized in Figure 1.

### 6.1. Checkpoint Therapy

As discussed above in Section 3, anti-PD-1/anti-PD-L1 monoclonal antibodies are currently being used in clinical trials to augment NK cell activity in NK cell adoptive therapy. However, to our knowledge, checkpoint therapies that target NK cell inhibitory receptors or their ligands beyond the PD-1/PD-L1 axis have not yet been tested in clinical trials as a combination therapy for adoptively-transferred NK cells. A promising novel target for NK cell-directed checkpoint therapy is TIGIT, due to its evident role in suppressing NK cell anti-tumor activity [140]. In murine lung metastasis models, it was shown that IL-15 therapy was more effective in reducing tumor burden when infused with TIGIT-deficient NK cells compared to WT NK cells [141], thus providing evidence that TIGIT blockers, combined with IL-15 therapy and NK cell adoptive transfer, could be an interesting therapeutic strategy. In addition, the use of antibodies that block NK cell TIM-3, LAG-3, KIRs, NKG2A, or Siglec-7 or -9, or their ligands, has been described in an excellent recent review [142], and their use alongside adoptive NK cell transfer merits investigation.

The efficacy of combining checkpoint blockade with other inhibitors or drugs to improve NK cell tumor clearance is beginning to be explored, and could be applicable to adoptive NK cell transfer. For example, pharmacological inhibition of Vps34, a kinase involved in autophagy, causes the microenvironment of melanoma and colorectal cancer tumors to become more pro-inflammatory, leading to enhanced recruitment of NK cells and other immune cells, and ultimately more effective anti-PD-1/PD-L1 therapy [143]. An IL-15 superagonist, named N-803 (formerly ALT-803), which has seen widespread success as an immunotherapeutic option for multiple cancers [144,145], is being tested in a phase II clinical trial (NCT03853317) in combination with adoptive CAR-NK cells targeting PD-L1 after promising results in pre-clinical studies [47]. Interestingly, a novel molecule comprising N-803 and anti-PD-L1 domains has shown promise for different carcinoma models [144]. Another study showed that a CDK8/CDK19 inhibitor, which augments NK cell activity, improved anti-PD-1 treatment in the tumor model with MC38 colon adenocarcinoma cells [146]. Beyond PD-1, a combination of anti-TIGIT or anti-KLRG1 antibodies and DNA methyltransferase inhibitor reversed the metastasis-promoting state acquired by certain tumor-exposed NK cells [147].

Another potential combination therapy is the use of checkpoint molecules with suppressor cell depletion or inhibition. Suppressor cells, such as tumor-associated neutrophils and MDSCs, represent a major obstacle for optimal NK cell function in the tumor microenvironment, and are therefore prime targets for immunotherapy. Tumor-associated neutrophils repress NK cell cytotoxicity through the PD-1/PD-L1 axis, and their depletion combined with PD-L1 blockade was significantly better in reducing tumor burden than either monotherapy [46]. Moreover, elimination of MDSCs with PD-L1-directed CAR-NK cells was a highly effective therapy in a MOC1 murine tumor model when combined with an anti-PD-1 antibody and the IL-15 super-agonist N-803 [47].

### 6.2. Antibody-Based Constructs

Antibody-based constructs, that crosslink NK cell and tumor cell receptors have been developed primarily to improve NK cell recruitment, although they often have additional functions, such as enhancing NK cell activation. AFM13, a tetravalent chimeric antibody that targets CD30 and CD16A, has previously shown efficacy for Hodgkin’s lymphoma [148,149], and is currently being tested in a phase I clinical trial for R/R CD30+ lymphoid malignancies (NCT04074746). Bispecific or trispecific cell engagers (BiKEs or TriKEs), which are synthetic molecules that contain single-chain variable fragments (scFv) to one (BiKE) or two (TriKE) tumor antigens and an NK cell receptor, often CD16, represent another approach. A recent upgrade to TriKEs is the replacement of the second tumor antigen-targeting scFv with IL15, as a way of increasing NK cell activation and expansion, simultaneously to promoting NK cell trafficking to tumors [150]. An anti-CD16 ×IL-15 × anti-CD33 TriKE is currently being tested in a phase I/II clinical trial (NCT03214666), and a second generation version has already been developed [151]. Other advancements include the development of: (i) a BIKE that additionally includes an anti-CD133 molecule to target cancer stem cells [152]; (ii) a TriKE that activates NK cells by binding NKp46 instead of presenting IL-15 [153]; and (iii) a nanoparticle-based TriKE that targets two NK cell activating receptors (CD16 and 4-1BB) and the tumor antigen EGFR, while simultaneously delivering the chemotherapeutic epirubicin [154]. Future potential modifications to TriKEs are numerous and could include blocking NK inhibitory receptors or, by using the aforementioned nanoparticles, delivering activity-boosting drugs, metabolism-fueling nutrients, such as glucose or essential amino acids, or STING agonists, which have recently been shown to enhance NK cell-mediated tumor rejection [155].

### 6.3. Genetic Modification

Beyond the transduction of a CAR, adoptively-transferred NK cells can be genetically manipulated in other ways to boost their effectiveness. Inducing the expression of chemokine receptors on CAR-NK cells can dramatically improve tumor homing, as demonstrated for CXCR4 and CXCR1, co-expressed on EGFRvIII-specific or NKG2D ligand-targeting CAR-NK cells, respectively [156,157]. Infiltration of adoptively-transferred conventional NK cells into solid tumors is also augmented by induced expression of chemokine receptors [158,159].

On the other hand, NK cells can be genetically modified to enhance their cytolytic activity. For example, NK cells can be transduced with the dominant negative TGF-β receptor II to block inhibitory TGF-β signaling [160]. Another approach is to transduce the genes encoding IL-2 or IL-15 to enable self-production of these critical cytokines [161]. Similarly, genetic deletion of a negative regulator of IL-15 signaling, called cytokine-inducible SH2-containing protein or CIS, in iPSC-NK cells led to improved anti-tumor function, both in vitro and in vivo in a leukemia xenograft model [162]. The authors of this study showed that the CIS-deficient NK cells had improved metabolic fitness, caused by upregulated mTOR activity, suggesting that the mTOR inhibitors that are currently being trialed for various cancers may inadvertently dampen NK cell metabolic activity and therefore function. Moreover, genetic deletion of CIS protein in “armored” IL-15-secreting anti-CD19 CAR-NK cells boosted their activity, and did not induce any signs of measurable toxicity in lymphoma xenografts [163]. A current limit to the genetic manipulation of primary NK cells is the difficulty of achieving high transduction efficiency, and methods to overcome this are likely to greatly expand the therapeutic genetic engineering of adoptively transferred NK cells.

### 6.4. Memory-Like NK Cells

A pertinent major challenge with NK and CAR-NK cell adoptive therapy is the long-term persistence of NK cell activity in the host. As previously mentioned, one potential strategy to increase the survival and functionality of transferred NK cells is to induce a memory-like phenotype in the NK cells ex vivo prior to transfusion by pre-activating NK cells with IL-15, IL-12, and IL-18 [25,164]. Clinical trials are currently ongoing to assess the efficacy of these memory-like NK cells to treat AML and/or MDS (NCT01898793, NCT02782546, NCT03068819, NCT04024761, NCT04354025) [25], and head and neck squamous cell carcinoma (NCT04290546). Using memory-like NK cells to generate CAR-NK cells is an exciting prospect to improve CAR-NK therapy, as demonstrated in a recent study which showed that memory-like CD19-directed CAR-NK cells display improved functional responses, compared to conventionally generated CAR-NK cells [165].

Less differentiated NK cells which display a memory-like phenotype can also be generated by ex vivo expansion using 721,221 feeder cells with membrane-bound IL-21 [166]. Critically, these cells upregulated genes related to glucose and amino acid metabolism, including amino acid transporter SLC7A5 and the transcription factor c-Myc, which are critical for NK cell function [167]. Interestingly, glucose metabolism is vital for memory-like NK cell tumor-clearing ability [168]. Due to the nutrient-deficient nature of the tumor microenvironment, metabolic rewiring of NK cells is likely to be key in their survival and persistence of their function. Future NK cell-specific CAR engineering should consider the engagement of signaling pathways that sustain NK cell metabolism.

### 6.5. Preventing NK Cell Senescence

The development of dysfunction is another barrier to NK cell persistence, and senescence may be partly responsible. Long-term ex vivo expansion of NK cells can cause replicative senescence induction [169]. This can be alleviated by using artificial antigen presenting cells (aAPCs) expressing membrane-bound IL-21 in place of aAPCs that express membrane-bound IL-15 [170], or transduction of the gene encoding the catalytic subunit of human telomerase, hTERT [169]. In cancer, NK cell senescence has not been clearly defined [171], but NK cells do release SASP-like molecules in response to tumor cells, and upregulate KLRG1 and ATM phosphorylation, key senescence markers, in response to chronic stimulation [106,172]. Furthermore, blocking the ATM-dependent DNA damage response pathway during NK cell activation led to enhanced NK cell activity, downregulation of KLRG1, improved persistence, and better tumor clearance in a murine B cell lymphoma model [172]. ATM inhibitors represent an attractive way of preventing NK cell senescence, and could be used, either during the ex vivo expansion of therapeutic NK cells, or as a treatment given to patients following NK cell infusion. Although there are currently no FDA-approved ATM inhibitors, many are being assessed as cancer therapeutics in clinical trials [173]. Overall, future work to further investigate how NK cell senescence affects NK cell adoptive transfer therapy would be of great benefit.

## 7. Conclusions

In summary, recent clinical studies of administered NK cells have not demonstrated a high efficacy in cancer patients. New approaches that combine NK cells with monoclonal antibodies or other drugs have shown increased efficacy in pre-clinical models, but critically, are yet to be demonstrated in patients. Genetically-modified NK cells, such as CAR-NK cells, have opened a new pathway to improving the efficacy of NK cell therapy. Other genetic modifications could be implemented in NK cells to improve homing to tumor sites, enhance NK cell activation, remove NK cell suppression, and improve the persistence of NK cells. Moreover, studies in microbial infections have shown that NK cells have an important immunoregulatory role for T cells, including the elimination of pathophysiological T cells, and that each immune cell subset has a specific timing for their action. Therefore, an efficient immune response requires a balance between both T cells and NK cells, and this should be considered when designing an efficient immunotherapy strategy.

## Figures and Tables

**Figure 1 cancers-12-03139-f001:**
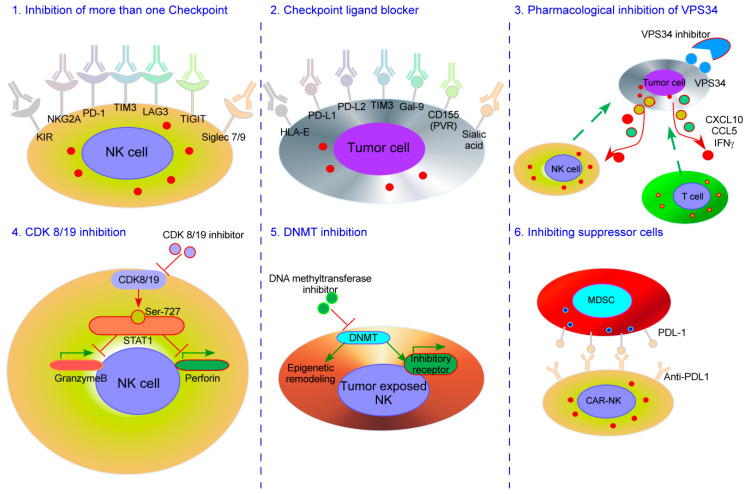
Novel strategies to enhance NK cell checkpoint inhibitors. NK: natural killer; KIR: Killer-cell immunoglobulin-like receptor; DNMT: DNA methyltransferase; Ser: serine; MDSC: myeloid-derived suppressor cell.

**Table 2 cancers-12-03139-t002:** Clinical results that have administered NK cells in combination with antibodies within the last 5 years.

Study Phase. NCT Number. Reference	Monoclonal Antibody	Source of NK and Method of Expansion	Condition (Disease) and Number of Patients	Clinical Outcome
Phase I/II NCT02030561 Lee et al. [32]	Trastuzumab (anti-HER2)	Autologous PBNK + K562-mb15-41BBL + IL-2 for 10 days	HER2^+^ refractory cancer. Phase I (*n* = 9)/II (*n* = 20)	Phase I: well tolerated. 66.6% SD (>6 months). 11% PR.
Phase I/II NCT02845856 [33]	Cetuximab (anti-EGFR)	Allogeneic PB-NK + K562-mb15-41BBL	Non-small Cell Lung Cancer (*n* = 54)	OS 9.5 months and PFS 6 month vs. 7.5 and 4.5 given Cetuximab alone
Phase I/II NCT02843204 Lin, M. et al. [44]	Pembrolizumab (anti-PD1)	Allogenic PB-NK + 10 IU/mL IL-2	Advanced non-small cell lung cancer (*n* = 109)	OS 15.5 months and PFS 6.5 months vs. 13.3 and 4.3 given Pembrolizumab alone
Phase II NCT03958097 Hospital of Jilin University	Sintilimab (anti-PD1)	Autologous PB-NK	Non-small cell lung cancer	NRP
Phase I NCT03841110	Nivolumab (anti-PD1), Pembrolizumab (anti-PD1), Atezolizumab (anti-PDL1)	iPSC-derived NK cell (FT500)	Advanced solid tumors and lymphomas (*n* = 76 estimated)	NRP
Phase I NCT03815084	Pembrolizumab (anti-PD1)	DC and NK cells	Solid tumors	NRP
Phase I/IIa NCT03937895	Pembrolizumab (anti-PD1)	Allogeneic NK Cell (“SMT-NK”)	Advanced Biliary Tract Cancer	NRP
Phase II NCT03853317	Avelumab (anti-PDL1)	Off-the-shelf CD16-targeted NK cells with intracellular IL2 (haNK) and IL-15 Superagonist (N-803)	Merkel Cell Carcinoma	NRP

SD: stable disease; PR: partial response; OS: overall survival; PFS: progression free survival; ORR: overall response rate; NRP: no results posted; DC: dendritic cell; PB-NK: peripheral blood NK cells.

**Table 3 cancers-12-03139-t003:** Pre-clinical studies with chimeric antigen receptor (CAR)-modified NK (CAR-NK) cells obtained from NK cell lines.

NK source	Target	Disease	Efficacy	Reference
NK92	GD2	NB	Higher in vitro efficacy against NB cell lines and primary NB cells.	Esser, R. et al. [56]
NK92	CD5	T-ALL	Higher in vitro efficacy against T-ALL and lymphoma cell lines, primary cells and in vivo in models of T-ALL.	Chen, K.H. et al. [57]
NK92	CD20/CD19	B-ALL	Higher in vitro efficacy against primary CLL cells than NK-92 combined with anti-CD20 Ab (rituximab or ofatumumab). Higher in vivo efficacy in models of B-ALL. CAR anti-CD20 was better than CAR anti-CD19.	Boissel, L. et al. [58]
NK92	ERbB2	GBM	Higher in vitro and in vivo efficacy of CAR-NK vs. parental NK-92 in murine models of GBM	Zhang, C. et al. [59]
NK92	EGFR/EGFRvIII	GBM	CAR targeted both wtEGFR and EGFRvIII and displayed enhanced in vitro and in vivo efficacy in a EGFR-dependent manner.	Han, J. et al. [60]
NK92	HER2	Breast cancer/ Renal carcinoma	Enhanced in vivo efficacy in orthotopic breast carcinoma xenografts, and reduction of pulmonary metastasis in renal carcinoma model.	Schönfeld, K. et al. [61]
NK92	CS1	MM	Enhanced in vitro and in vivo efficacy in aggressive orthotopic MM xenograft mouse model.	Chu, J. et al. [62]
NK92	CD138	MM	Enhanced specific cytotoxicity against MM cell lines and primary MM cells, and in a xenograft NOD-SCID mouse MM model.	Jiang, H. et al. [63]
YTS	PSCA-DAP12	Prostate tumor	CAR with DAP12 improved in vivo efficacy compared to CARs with CD3z signaling domain in murine models.	Töpfer, K. et al. [64]
KHYG-1	EGFRvIII	GBM	Specific enhanced in vitro efficacy.	Murakami, T. et al. [65]
NK92	Anti-αFR	Ovarian cancer	1st, 2nd (CD28) and 3rd (CD28 and 4-1BB) generation CARs were compared. Higher efficacy of 3rd generation CAR in vitro and in mouse models.	Ao, X. et al. [66]
NK92	PD1-NKG2D	Solid Tumor	Comparison of different CARs with PD1 and NKG2D and different transmembrane domains. CAR with NKG2D hinge region and 4-1BB co-stimulatory domain presented enhanced in vitro efficacy.	Guo, C. et al. [67]
NK92	2, 4-dinitrophenyl	HIV	CAR targets epitopes of gp160. Eliminates HIV-infected primary CD4+ T cells.	Lim, R. et al. [68]

NB: Neuroblastoma; T-ALL: T cell acute lymphoblastic leukemia; B-ALL: B- acute lymphoblastic leukemia; CLL: chronic lymphocytic leukemia; GBM: Glioblastoma; MM: multiple myeloma.

**Table 4 cancers-12-03139-t004:** Pre-clinical studies with CAR-NK cells obtained from primary NK cells.

NK Source	Target	Disease	Efficacy	Reference
PB	CD19/GD2	Leukemia/NB	CARs incorporating 2B4 signaling into CD3z show enhanced in vitro efficacy against leukemia or NB cells.	Altaver et al. [76]
PB	CD20	BL	CAR-NK cells were obtained by electroporation and showed enhanced in vitro and in vivo efficacy in models of BL receiving multiple doses of CAR-NK.	Chu et al. [77]
PB	CD19	B-cell malignancies	Expansion with K562-based antigen presenting cells with membrane bound IL-15 yielded a median greater than 1000-fold expansion of CAR-NK cells at 3 weeks of culture. Enhanced NK-cell-mediated killing of leukemic cells with 4-1BB incorporation in the CAR construct.	Imai et al. [70]
PB	HER-2	Ovarian cancer	Specific in vitro and in vivo response which correlated with level of HER-2 on tumor cells.	Kruschinski et al. [78]
PB	CD19	B-ALL and CLL	NK cells were modified based on transfection of mRNA which persisted up to 3 days showing in vitro efficacy.	Li et al. [79]
PB	CD19	Leukemia	NK cells were modified by electroporation with the corresponding mRNA. CAR expression reached maximum levels of expression at 24–48 h; specific killing was observed at 96 h.	Shimasaki, et al. [80]
CB	CD19	NHL	IL-15 in the construct increases persistence and in vivo efficacy in a NHL murine model. iCAS-9 in the CAR construct. Presence of CAR-NK cells up to 68 days.	Liu et al. [23]
iPSC	hMesothelin	Ovarian cancer	CAR with NKG2D transmembrane and 2B4 co-stimulatory domains presents high specific NK cell efficacy in an ovarian cancer xenograft model, showing similar in vivo activity to CAR-T cells, although with less toxicity.	Li et al. [74]

PB: peripheral blood; CB: cord blood; iPSC: induced pluripotent stem cells. NB: neuroblastoma; NHL: non-Hodgkin-lymphoma; BL: Burkitt lymphoma; B-ALL: B cell acute lymphoblastic leukemia; CLL: chronic lymphocytic leukemia; iCAS-9: inducible Caspase-9.

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
