# Peer review of "Natural Killer Cells in Immunotherapy: Are We Nearly There?"

_cancers, 2020, doi:10.3390/cancers12113139_

Round 1

Reviewer 1 Report

Authors have done an exceptional job of summarizing the recent progress in the therapeutic applications of NK cells. Well-written and articulating the strength and limitations of utilizing NK cells in the clinic. Authors should include a cartoon describing different methods listed in their manuscript on how to improve the genetically-modified NK therapy.

Author Response

REVIEWERS’ COMMENTS AND OUR REPLIES:

Reviewer #1 (Remarks to the Author):

Comments and Suggestions for Authors

Authors have done an exceptional job of summarizing the recent progress in the therapeutic applications of NK cells. Well-written and articulating the strength and limitations of utilizing NK cells in the clinic. Authors should include a cartoon describing different methods listed in their manuscript on how to improve the genetically-modified NK therapy.

  • We thank the reviewer for all the positive comments. We have included a new Figure (Figure 1) in the manuscript.

Reviewer 2 Report

The review proposed by Mireia Bachiller and her collaborators is divided in two separate parts addressing different topics: NK cell based immunotherapies against cancer and anti-microbial NK cell responses.

For the first part on cancer, numerous reviews have been proposed on NK cell immunotherapy in cancer. Mainly, Guillerey et al. (Nature Immunol. 2016) proposed a well-illustrated overview including table summarizing all NK cell based trials. Last year, Hodgins et al (JCI 2019) proposed a similar review with exhaustive table including all trials based on CAR-NK cells.

Mireia Bachiller et al. proposed a long and exhaustive review on the NK cell based immunotherapies against cancer. Four tables grouping trials and studies illustrate the review’s content. It is a faithful review of the ongoing NK cell immunotherapies in cancer. However, as numerous reviews co-exist in this field, authors should propose an attractive structuration to catch reader’s interest. The authors should modify the structuration to improve the readability of this part. Illustrations should be appreciable, particularly for the part 5.

The second part of the review is focused on NK cell responses against microbial infections. It is difficult to propose a review like this in which two issues are treated differently. Concerning the second part on anti-microbial NK cell responses, the link is inexistent.

Overall, the authors have to modify significantly the structure of this review with a common thread or to propose 2 different reviews with a clear message.  

Comments addressed to the authors:

line 31: add the reference Cooper MA et al. Trends Immunol 2001

Line 48: add “T-depleted haploidentical” HSCT

Table 2 typing error for KHYG-1 “specific enhanced in vitro efficacy”

Wrong names in references 3, 4 and 101

The following references should be added:

  • Daher Blood 2020
  • Guillerey et al. Nature Immunol. 2016
  • Hodgins et al JCI 2019

Although the structuration of part 5 following 4 themes (targeting, enhancing activation, removing suppression and persistence/survival) should be attractive, it is complicated to bring a clear view of all approaches as immunotherapeutic strategies cannot be regarded as only one theme. For example, lines 294 to 301: Bike and trike have been designed to promote the targeting of NK to tumor cells. They are also presented in the following paragraph dedicated to “enhancing activation”. It is a mix difficult to follow.

Line 326, add the reference of Salter et al. Sci Signal 2018.

Line 394, the authors should modulate their message as NKG2C+ NK expansion is observed only in a few proportion of CMV+ individuals.

Line 398, the authors reference Petitdemange et al (105) and conclude that CHIKV contributes to NKG2C+ NK cell expansion. However, in the reference 105 in which Petitdemange et al described the NK cell phenotype of CHIKV positive patients, all patients were CMV+ Africans. Thus the authors cannot exclude that NKG2C+ expansion is due to CMV.

The part 6.1 “Virus” should be ended line 426. A title should be added for the following paragraph which is focused on immune responses against virus.

Author Response

Reviewer #2 (Remarks to the Author):

The review proposed by Mireia Bachiller and her collaborators is divided in two separate parts addressing different topics: NK cell based immunotherapies against cancer and anti-microbial NK cell responses. For the first part on cancer, numerous reviews have been proposed on NK cell immunotherapy in cancer. Mainly, Guillerey et al. (Nature Immunol. 2016) proposed a well-illustrated overview including table summarizing all NK cell based trials. Last year, Hodgins et al (JCI 2019) proposed a similar review with exhaustive table including all trials based on CAR-NK cells. Mireia Bachiller et al. proposed a long and exhaustive review on the NK cell based immunotherapies against cancer. Four tables grouping trials and studies illustrate the review’s content. It is a faithful review of the ongoing NK cell immunotherapies in cancer. However, as numerous reviews co-exist in this field, authors should propose an attractive structuration to catch reader’s interest. The authors should modify the structuration to improve the readability of this part. Illustrations should be appreciable, particularly for the part 5.

  • We thank the reviewer for all the positive comments. We agree with the reviewer on re-structuring the manuscript to catch the reader’s interest. We have re-structured the manuscript, especially previous sections 5 and 6. We have added a figure (Figure 1) regarding previous section 5. In addition, we agree with the reviewer that there have been some excellent reviews published on NK cell immunotherapy, and we believe that because our review focusses on clinical trials and pre-clinical studies from the last 5 years, we provide important updates on what is a rapidly evolving topic. Furthermore, we believe that in sections 5 & 6 we highlight some novel ideas about how to improve NK cell immunotherapy that have not previously been discussed together with NK cells and CAR-NK cells in an exhaustive review.

The second part of the review is focused on NK cell responses against microbial infections. It is difficult to propose a review like this in which two issues are treated differently. Concerning the second part on anti-microbial NK cell responses, the link is inexistent. Overall, the authors have to modify significantly the structure of this review with a common thread or to propose 2 different reviews with a clear message.  

  • We have rewritten the previous section 6 (now section 5): “Antimicrobial properties of NK cells” to give a stronger link to the rest of the manuscript and catch the reader’s attention. The new section 5 is now termed: “LESSONS LEARNED FROM THE ANTIMICROBIAL PROPERTIES OF NK CELLS”.

Specific comments addressed to the authors:

  1. Line 31: add the reference Cooper MA et al. Trends Immunol 2001.

  • We thank the reviewer for this suggestion and we have added that reference (new reference 1).

  1. In section 1d: Line 48: add “T-depleted haploidentical” HSCT.

  • We have added that information (Section 1, line 53).

  1. Table 2 typing error for KHYG-1 “specific enhanced in vitro efficacy”

  • We have modified that mistake in Table 3.

  1. Wrong names in references 3, 4 and 101

  • Thank you for noticing that. We have modified them.

  1. The following references should be added:

  • Daher et al. Blood 2020: this reference has been added in new section 6.3.
  • Guillerey et al. Nature Immunol. 2016; This reference has been added in the introduction to section 6.
  • Hodgins et al. JCI 2019: this reference has been added in the introduction to section 6.

  1. Although the structuration of part 5 following 4 themes (targeting, enhancing activation, removing suppression and persistence/survival) should be attractive, it is complicated to bring a clear view of all approaches as immunotherapeutic strategies cannot be regarded as only one theme. For example, lines 294 to 301: Bike and trike have been designed to promote the targeting of NK to tumor cells. They are also presented in the following paragraph dedicated to “enhancing activation”. It is a mix difficult to follow.

  • We agree with the reviewer and now section 6 (previous section 5) has been re-structured in 4 new sections: 6.1: Checkpoint therapy; 6.2: Antibody-based constructs; 6.3: Genetic modification; 6.4: Memory-Like NK cells.

  1. Line 326, add the reference of Salter et al. Sci Signal 2018.

  • As the structure of section 5 has new changes, that reference has been added in section 4.

  1. Line 394, the authors should modulate their message as NKG2C+ NK expansion is observed only in a few proportion of CMV+ individuals

  • We have modified the sentence and now it says: “Afterwards, this event was also observed in humans during CMV infection, where a higher expansion of NK cells expressing NKG2C and CD57 was detected in a small proportion of CMV+ individuals”

  1. Line 398, the authors reference Petitdemange et al (105) and conclude that CHIKV contributes to NKG2C+ NK cell expansion. However, in the reference 105 in which Petitdemange et al described the NK cell phenotype of CHIKV positive patients, all patients were CMV+ Africans. Thus, the authors cannot exclude that NKG2C+ expansion is due to CMV.

  • The reviewer is right, the authors of that reference cannot exclude that the expansion could be due to CMV. We have modified that sentence and now it states: “The existence of memory-like NK cell development after viral infection was confirmed for other viruses, such as Chikungunya virus, which also induces a preferential expansion of NKG2C+ NK cells. However, at this study, all individuals were CMV+; thus, it cannot be excluded that this preferential expansion was due to CMV”.

  1. The part 6.1 “Virus” should be ended line 426. A title should be added for the following paragraph which is focused on immune responses against virus

  • We have re-structured that section (new section 5) in these sub-sections: 5.1. NK cell responses and development of memory-like NK cells after viral infections; 5.2. Coordinated immune response after viral infections; 5.3. Collaboration of NK cells and T cells after parasites infections; 5.4: Coordinated immune response against bacteria for NK and T cells; 5.5. NK cells alone and in collaboration to fight fungal infections.

Reviewer 3 Report

This review examines a very current and very interesting topic. It is well organized and comprehensively described. The work includes some appropriate references, however further references need to be included and discussed. In particular, in sections 2 and 3, the authors should discuss immunotherapeutic approaches based on the use of NK cell immune checkpoint inhibitors, such as monalizumab and lirilumab, in both solid and haematological tumors. They should also include work describing PD-1 expression and additional immune checkpoints (i.e. NKG2A, KIR) on NK cells from healthy and cancer patients to better clarify the rationale for the use of anti-PD-1 mAbs plus anti-NKG2A and / or anti-KIR mAbs as additional immunotherapeutic approaches in different conditions (e.g. Andre A et al, Cell 2018; Pesce S et al JACI 2017, Roberto A et al, Haematologica 2018; ...). Finally, in section 6, authors should include and discuss articles on direct / indirect interactions between NK cells and BCG (Esin S et al, Immunology 2004; Marcenaro E et al, Int Immunol 2008; Esin S et al, Inf and Immun 2008 ; ..).

Author Response

Reviewer #3 (Remarks to the Author):

  1. This review examines a very current and very interesting topic. It is well organized and comprehensively described. The work includes some appropriate references, however further references need to be included and discussed. In particular, in sections 2 and 3, the authors should discuss immunotherapeutic approaches based on the use of NK cell immune checkpoint inhibitors, such as monalizumab and lirilumab, in both solid and haematological tumors.

  • We thank the reviewer for all the positive comments. Regarding section 2, we mention results in the last 5 years, as in 2017 we did an extensive review where we describe studies administering NK cells alone, and now we have included additional new studies in the last 5 years. Regarding Section 3, we have added additional references including information related to studies combining NK cells with monalizumab and lirilumab. We recognize that these studies are specifically important in the field of NK cell immunotherapy and merit inclusion in our review. Beyond these studies, we do not exhaustively detail every study that has been conducted using immune checkpoint inhibitors as others have already done an excellent job in doing this (for example, Zhang et al, Frontiers in Immunology, 2020) and we instead focused on those that have combined immune checkpoint inhibitors with NK cell therapy.

  1. They should also include work describing PD-1 expression and additional immune checkpoints (i.e. NKG2A, KIR) on NK cells from healthy and cancer patients to better clarify the rationale for the use of anti-PD-1 mAbs plus anti-NKG2A and / or anti-KIR mAbs as additional immunotherapeutic approaches in different conditions (e.g. Andre A et al, Cell 2018; Pesce S et al JACI 2017, Roberto A et al, Haematologica 2018; ...).

  • We have added additional information and references regarding the expression of these checkpoints in NK cells and other pre-clinical studies. We also include these references (Andre A et al, Cell 2018; Pesce S et al JACI 2017, Roberto A et al, Haematologica 2018) in section 3.

  1. Finally, in section 6, authors should include and discuss articles on direct / indirect interactions between NK cells and BCG (Esin S et al, Immunology 2004; Marcenaro E et al, Int Immunol 2008; Esin S et al, Inf and Immun 2008 ; ..).

  • We thank the reviewer for this suggestion as we were not aware of it. In section 5 (previous section 6), we have added information related to interactions between NK cells and Mycobacterium bovis bacille Calmette-Guérin including references suggested by the reviewer.

Round 2

Reviewer 2 Report

I would like to congratulate the authors for improving their review. They have considered all suggestions. It is an exhaustive and insightful review.